# Histone Modifications: Potential Therapeutic Targets for Diabetic Retinopathy

**DOI:** 10.3390/biom15040575

**Published:** 2025-04-12

**Authors:** Yao Lu, Yizheng Zhang, Jin Yao, Wen Bai, Keran Li

**Affiliations:** 1Department of Ophthalmology, The Affiliated Eye Hospital of Nanjing Medical University, Nanjing 210029, China; 2023121343@stu.njmu.edu.cn (Y.L.); yaojin@njmu.edu.cn (J.Y.); 2The Fourth School of Clinical Medicine, Nanjing Medical University, Nanjing 210029, China; zhangyizheng@stu.njmu.edu.cn

**Keywords:** histone modification, diabetic retinopathy, epigenetic, epigenetic therapy

## Abstract

Diabetic retinopathy (DR) is a microvascular complication arising as a secondary effect of diabetes, with both genetic and environmental factors playing a significant role in its onset and progression. Epigenetics serves as the crucial link between these genetic and environmental influences. Among the various epigenetic mechanisms, histone modification stands out as a key regulatory process associated with the development of many diseases. Histone modifications primarily regulate cellular function by influencing gene expression. Modulating histone modifications, particularly through the regulation of enzymes involved in these processes, holds a promising therapeutic approach for managing diseases like DR. In this review, we explore the regulatory mechanisms of histone modification and its contribution to the pathogenesis of DR.

## 1. Introduction

Diabetic retinopathy (DR) is a common and chronic complication of diabetes, particularly affecting individuals over 40, and is a leading cause of blindness in this group [1]. In the early stages, hyperglycemia damages the vascular endothelium, causing leakage that manifests as microaneurysms, hemorrhages, and exudates [2]. As DR progresses, increased ischemia leads to the formation of fragile new blood vessels prone to rupture, resulting in vitreous hemorrhage and retinal detachment [3,4]. Chronic inflammation and neurodegenerative changes gradually destroy retinal tissue, eventually causing severe vision loss [5]. Current treatments, including intravitreal steroid injections, laser photocoagulation, and anti-vascular endothelial growth factor (VEGF) monoclonal antibodies (e.g., bevacizumab), can be effective but are limited by side effects. This underscores the need for safer, more effective therapies. Epigenetic therapy, which regulates gene expression without altering the DNA sequence, is emerging as a promising strategy for DR. Epigenetic modifications play a crucial role in regulating genes involved in angiogenesis, inflammation, and neurodegeneration, making therapeutic approaches targeting these modifications a potential breakthrough in DR treatment.

Epigenetic modifications, including post-translational histone modifications, DNA methylation, and N6-methyladenosine modifications, do not alter the DNA sequence but can affect gene expression [6]. Hyperglycemia influences the transcription and expression of genes by regulating various epigenetic modifications, playing a key role in DR. Recently, histone modifications have gained particular attention, as these changes, catalyzed by specific enzymes, occur at the amino-terminal ends of histones and impact fundamental biological processes like replication, transcription, and chromosome maintenance [7]. Modifications such as histone acetylation [8,9], methylation [10], ubiquitination [11], phosphorylation [12], and citrullination, which are influenced by high glucose metabolism, contribute to the pathological mechanisms of DR. This article focuses on the progress of research on these modifications in DR, highlighting their roles and potential for therapeutic intervention.

## 2. Overview of Histone Modification

### 2.1. Composition and Structure of Histones

The histone family consists of diverse proteins classified into nucleosomal and non-nucleosomal histones. Nucleosomal histones, including H1, H2A, H2B, H3, and H4, play a crucial role in DNA packaging and the formation of nucleosome structures [13,14]. These proteins can be further divided based on their amino acid composition: lysine-rich histones (H1, H2A, H2B) and arginine-rich histones (H3, H4) [15]. H2A, H2B, H3, and H4, known as core histones, form octamers that constitute the nucleosome core, while H1, the linker histone, stabilizes the nucleosome structure by binding outside the core [16,17]. Non-nucleosomal histones do not form nucleosome structures but function in other regions of the nucleus [18,19]. This review focuses on the significance of nucleosomal histone modifications in the context of DR.

### 2.2. Biological Functions of Histone Modifications

Histone modifications involve the addition of specific chemical groups or small proteins, such as ubiquitin, to histone amino acid residues, playing a critical role in regulating chromatin structure and gene expression (Figure 1). These modifications can alter the affinity between histones and the DNA double helix, causing chromatin to either relax or condense [20]. They also influence the binding of transcription factors to gene promoters, modulating gene expression and maintaining genomic stability [21].

Histone modifications regulate chromatin structure by affecting its openness. Chromatin consists of euchromatin (open) and heterochromatin (closed), which determines gene activation or repression [22]. Modifications that add negative charges to histones weaken their electrostatic interactions with the negatively charged DNA, relaxing chromatin into an open euchromatin state [23]. This increases DNA accessibility, promoting transcription factor binding and gene activation. In contrast, heterochromatin, associated with gene silencing, is more condensed [24,25,26,27]. Histone modifications recruit proteins like heterochromatin protein 1 (HP1) and Polycomb repressive complex 1 (PRC1), promoting chromatin condensation and gene repression [28,29,30]. Removing negatively charged groups restores histone positivity, tightening chromatin and inhibiting transcription factor binding, leading to gene silencing.

Histone modifications are reversible covalent changes regulated by specific enzymes and cofactors. These cofactors are categorized as “writers”, “erasers”, and “readers” [31,32]. “Writers” catalyze the addition of chemical groups to histones, altering their charge and structure, thus regulating chromatin states. “Erasers” remove these modifications, reversing the effects of “writers”. “Readers” recognize specific modifications, recruiting chromatin remodeling complexes and transcription factors to translate these signals into biological responses, such as gene activation or repression. Through the coordinated actions of these enzymes, histone modifications precisely control chromatin structure and gene expression, playing a critical role in cellular function, development, and responses to environmental changes.

## 3. Histone Modifications and DR

Research has shown that histone modifications are integral to both physiological and pathological processes. Physiological events such as embryonic development, lactation, neural activity regulation, and metabolic functions all rely on histone modifications. Additionally, these modifications play key roles in pathological conditions like inflammation, immune imbalances, and fibrosis. In DR, a disease marked by inflammation and neurodegeneration, the progression is closely tied to the types and extent of histone modifications. A thorough investigation of how histone modifications affect DR can provide valuable insights into its underlying mechanisms and help identify more effective treatment strategies (Figure 2).

The inner circle of the diagram depicts the pathological events triggered by hyperglycemia—such as oxidative stress, inflammation, and angiogenesis—that drive the development of DR. The outer circle illustrates how histone modifications regulate gene expression, thereby influencing the progression of DR.

### 3.1. Histone Acetylation

Histone acetylation, a common modification occurring on lysine residues in the N-terminal regions of core histones, involves the transfer of a negatively charged acetyl group to the positively charged NH3^+^ group [33]. This weakens the interaction between histones and DNA, leading to chromatin relaxation [34]. Acetylation loosens the chromatin structure, allowing transcription factors to bind and promoting gene expression, while deacetylation tightens chromatin and suppresses transcription.

Histone acetylation is regulated by two main enzyme groups: histone acetyltransferases (HATs) and histone deacetylases (HDACs). HATs catalyze acetylation, typically linked to gene activation. They are categorized into families such as GNAT (e.g., GCN5, PCAF), MYST (e.g., MOZ, TIP60), p300/CBP (e.g., p300, CBP), and nuclear receptor coactivators (e.g., SRC-1, ACTR). HDACs remove acetyl groups, usually associated with gene repression, and are classified into four groups: Class I (HDAC1/2/3/8), Class II (HDAC4/5/6/7/9/10), Class III (SIRT1-SIRT7), and Class IV (HDAC11).

HDACs have complex roles in DR, exhibiting both protective and damaging effects. High glucose environments significantly increase HDAC expression, reducing histone acetylation. For instance, SIRT1, a deacetylase, exerts antioxidant effects and, when overexpressed in retinal cells, can inhibit high-glucose-induced senescence markers such as p53, p21 and PAI-1 [35,36,37]. In contrast, other HDACs enhance oxidative stress in DR. Elevated HDAC2 levels in high glucose reduce H3K9 and H3K27 acetylation, suppressing MnSOD expression and increasing ROS, exacerbating oxidative stress in endothelial cells [38]. Inhibition of HDAC3 activity can activate Nrf2 signaling (by suppressing Keap1 synthesis and Nrf2-Nox4 association), improving endothelial cell dysfunction and protecting against diabetic damage [39]. Moreover, high-glucose-induced H3K9 acetylation facilitates p53 binding to the p66Shc promoter, activating p66Shc transcription [40]. This induces reactive oxygen species (ROS) production through its redox activity or the Rac1-Nox2 pathway, contributing to retinal microvascular damage.

HDACs also have dual effects on pathological angiogenesis in DR. HDAC4 reduces acetylation, promoting transcription factors like HIF-1α and Nrf2, which are linked to neovascularization [41,42]. HDAC9 plays a significant role in angiogenesis by inhibiting microRNA-17-92 transcription [43]. Conversely, HDAC1 reduces H3K9ac levels, repressing genes involved in endothelial progenitor cell activity and inhibiting vascular function [44,45].

The roles and regulatory mechanisms of different HDACs in DR vary, suggesting that targeted therapeutic strategies could provide new insights for treatment. Plant extracts like resveratrol activate SIRT1, reduce H3K9ac levels, and alleviate oxidative stress and neuroinflammation, offering neurovascular protection [46,47]. Ginsenoside Rd enhances SIRT1 and SIRT3 activity, reducing ROS production and preventing retinal pigment epithelial cell apoptosis [48,49]. Curcumin activates HATs (p300/CBP) and inhibits HDACs (HDAC2), downregulating pro-inflammatory genes like TNF-α and IL-6, thereby alleviating inflammation and fibrosis [50,51]. HDAC inhibitors show neuroprotective effects in neurodegenerative diseases and may similarly protect against neurodegenerative changes in DR. Further research into the regulatory mechanisms of histone acetylation in DR could provide more targeted and effective treatment strategies for diabetic microvascular complications.

### 3.2. Histone Methylation

Histone methylation is the process by which one or more methyl groups are added to specific amino acid residues on histones, predominantly occurring at lysine (K) or arginine (R) residues in the N-terminal regions of histones H3 and H4 [52,53]. Lysine residues can undergo mono-methylation (me1), di-methylation (me2), or tri-methylation (me3), while arginine residues may undergo mono-methylation (MMA), symmetric di-methylation (SDMA), or asymmetric di-methylation (ADMA) [54]. These modifications regulate gene expression by altering chromatin structure and recruiting specific protein complexes. The functions of these modifications are site-specific and depend on the degree of methylation.

Histone methylation is regulated by enzymes, primarily histone methyltransferases (HMTs) and histone demethylases (HDMs), which, respectively, add or remove methyl groups from specific lysine and arginine residues. HMTs are categorized into histone lysine methyltransferases (KMTs) and histone arginine methyltransferases (PRMTs), with KMTs further divided into SET domain-containing enzymes and DOT1L. Notable SET domain-containing enzymes include EZH2, SETD1A/B, SETD2, SUV39H1/2, and G9a/GLP. HDMs are classified into the LSD and JMJD families, responsible for demethylating histones.

High-glucose conditions significantly affect mitochondrial function by regulating histone methylation at the promoters of protective and damaging genes. In the retinas of diabetic patients, elevated H4K20me3 levels and reduced H3K4me1/H3K4me2 levels were observed at the Sod2 promoter, leading to decreased Sod2 expression and increased mitochondrial damage, contributing to DR [55,56]. In diabetic rat retinas, decreased H3K9me2 levels at the MMP-9 promoter promoted NF-κB recruitment and MMP-9 activation, leading to mitochondrial damage and capillary cell apoptosis [57]. Inhibition of the demethylase LSD1 was found to prevent MMP-9 activation and offer partial retinal protection [58]. Hyperglycemia also increases H3K9me3 levels and decreases H3K9me2 levels at the Rac1 promoter, activating Rac1 expression, which in turn increases ROS production through the Nox2 pathway, causing mitochondrial damage and retinal lesions [59,60]. Inhibition of Suv39H1 reduces glucose-induced Rac1 transcription, preserving mitochondrial function and slowing retinopathy progression [61]. In summary, the high-glucose environment finely regulates mitochondrial function through complex histone methylation modifications, contributing to DR onset and progression.

The role of histone methylation in oxidative stress, DNA damage, and repair is gaining increasing support. In DR, hypoxia/reoxygenation (H/R) injury plays a key role in disease pathogenesis [62,63]. Inhibition of SETD7 alleviates H/R injury by downregulating Keap1 and promoting Nrf2-mediated antioxidant signaling [64]. Further research reveals that SETD7 knockout reduces the binding of H3K4me-stimulated protein 1 (Sp1) to the Keap1 promoter, inhibiting Keap1 expression and thereby enhancing Nrf2 and antioxidant signaling [65]. DNA damage is a critical mechanism in DR pathogenesis, promoting cell apoptosis [66]. Elevated PRMT4 levels were detected in the retinal pigment epithelium of diabetic rats, which promote H3R17me2 modification and RPE cell death, potentially due to DNA damage [67,68]. Histone methyltransferases and demethylases play crucial roles in the DNA damage response. Various H3K4me3 demethylases are recruited to double-strand break (DSB) regions during DNA damage, inhibiting gene transcription and facilitating DNA repair factor binding [69]. The histone methyltransferase GLP catalyzes H4K16me1 during DNA damage, promoting the recruitment of the DNA repair protein 53BP1 and enhancing non-homologous end joining (NHEJ) repair [70]. Additionally, H3K36me2 and H3K36me3 recruit important DNA repair factors, such as yKu70 (Ku70) and Rfa1 (RPA1), promoting NHEJ and homologous recombination (HR) repair, providing a theoretical foundation for DNA damage repair in DR [71].

Therapeutic strategies targeting histone methylation hold promise for preventing DR progression. Minocycline, a tetracycline antibiotic, has been shown to reduce H4K20me1/2 levels, inhibiting diabetes-induced chronic inflammation and alleviating diabetes-induced DNA damage in rodent retinas [72]. Exenatide-4, in addition to reducing histone acetylation, improves diabetes progression by restoring H3K4me and H3K9me2 levels, thus protecting the microvasculature [73]. Beyond drug therapy, histone methylation modifications may also serve as biomarkers for early DR diagnosis. Detecting histone methylation status in blood or retinal tissue can help identify high-risk patients and enable timely intervention.

### 3.3. Histone Lactylation

Histone lactylation, a novel epigenetic modification, was first identified by Zhang et al. in 2019, particularly in macrophages [74]. This process involves the addition of a lactyl group to lysine residues on histones, a modification mediated by lactate. Lactate, a byproduct of glycolysis, is tightly linked to cellular metabolic states and also functions as a signaling molecule that regulates gene expression. Lactate is transported through the “lactate shuttle”, which modulates its levels both within cells and in the surrounding environment. In diabetes, hyperglycemia enhances glycolysis, leading to increased lactate production in the vitreous of patients with proliferative diabetic retinopathy (PDR) [75]. Histone lactylation is typically associated with gene activation, influencing gene expression by modifying chromatin structure or recruiting specific transcription factors, playing a crucial role in the development of DR.

Chronic inflammation is a key pathological mechanism in DR, and histone lactylation may modulate retinal inflammatory responses by regulating anti-inflammatory gene expression. Microglia exhibit two primary phenotypes: M1 microglia, which release pro-inflammatory factors, and M2 microglia, which secrete anti-inflammatory cytokines. Studies have shown that H3K18la mediates the dysregulation of M1/M2 microglia through the NF-κB signaling pathway, exacerbating neuroinflammatory damage [76,77]. Additionally, H4K12la accumulates at the promoter regions of NF-κB pathway genes (e.g., Ikbkb, Rela, and Relb), activating transcription and promoting inflammation [78]. Targeting the NF-κB pathway regulated by H3K18la and H4K12la may offer a new therapeutic strategy for DR treatment.

Histone lactylation plays a complex and significant role in diabetic microvascular complications. Research suggests that H3K18la regulates the anti-inflammatory and pro-angiogenic dual activities of monocyte-macrophages by facilitating reparative gene transcription [79]. Specifically, lactate promotes histone lactylation at H3K18la, upregulating FTO in endothelial cells (ECs), which directly promotes retinal neovascularization and mediates microglial activation and neurodegeneration [80]. In endothelial cells, a negative feedback loop involving H3K9la and HDAC2, driven by hyperglycemia, facilitates VEGF-induced angiogenesis. Inhibiting lactylation disrupts the H3K9la/HDAC2 feedback loop, effectively suppressing DR-related neovascularization [81]. Additionally, histone lactylation can induce fibroblast differentiation and promote TGF-β-induced fibrosis [82]. H4K12la levels are positively correlated with inflammation and fibrosis, suggesting that lactylation may also contribute to retinal fibrosis in DR by activating fibrosis-related genes [78]. Inhibitors of LDHA, which suppress lactate production, can significantly improve diabetic fibrotic lesions, offering a potential therapeutic approach for diabetic microvascular complications by altering histone lactylation levels [83]. In summary, histone lactylation plays dual roles in diabetic microvascular complications, promoting both angiogenesis and fibrosis.

Understanding the target genes of histone lactylation in DR and exploring its temporal and spatial specificity are crucial areas for future research on histone lactylation modifications in DR. Studies have shown that both lactylation and acetylation occur at the ε-amino group of lysine, as both are acylation modifications, meaning histone lactylation is also regulated by acetyltransferases and deacetylases [74,84]. Consequently, histone deacetylase (HDAC) inhibitors, such as vorinostat, belinostat, and trichostatin A (TSA), could be potential clinical treatments targeting histone lactylation. Additionally, lactate accumulation is essential for histone lactylation, and inhibiting lactylation by using monocarboxylate transporter 1 (MCT1) inhibitors to block lactate uptake, or pyruvate dehydrogenase (PDH) inhibitors to suppress intracellular lactate production, may also provide potential strategies for DR treatment. Advancements in these areas will pave the way for precision therapies for DR patients.

### 3.4. Histone Citrullination

Histone citrullination primarily occurs on arginine residues of histones H1, H2A, H3, and H4, a process catalyzed by peptidylarginine deiminases (PADs). The PAD family consists of five isozymes (PAD1–4 and PAD6), each with tissue-specific expression patterns. PAD1 and PAD3 are mainly expressed in epithelial tissues, where they regulate skin barrier function [85]. PAD2 is widely expressed across various tissues, including the central nervous system and macrophages, playing crucial roles in neural function and immune-inflammatory responses [86,87]. PAD4, predominantly expressed in inflammatory cells such as neutrophils and macrophages [88,89,90], catalyzes the citrullination of histones H3 and H4 and is involved in inflammation and immune responses.

Recent research has shown that serum or intraocular levels of histone H3 citrullination (citH3) can serve as important biomarkers for assessing the severity of PDR. citH3 is a key marker for neutrophil extracellular trap (NET) formation. PAD4-induced histone H3 citrullination (citH3) is closely associated with NETosis, a process where neutrophils release NETs, contributing to inflammation and tissue damage [91]. In diabetic patients, elevated levels of PAD4 and NETosis markers have been detected, with serum NET levels correlating positively with the extent of vascular pathology [92]. NETs activate inflammatory signaling pathways, including NF-κB, leading to the release of pro-inflammatory cytokines and exacerbating retinal inflammation [93]. High mobility group box 1 (HMGB1), present in NETs, further amplifies inflammatory responses through both paracrine and autocrine mechanisms [94,95,96]. In addition to promoting inflammation, PAD4-induced citH3 levels have been shown to positively correlate with HIF-1α expression and the extent of neovascularization [97]. Proteolytic enzymes released by NETs can damage retinal vascular endothelial cells, contribute to thrombosis, and facilitate pathological neovascularization, thereby driving the progression of DR [98,99]. Interestingly, studies have also shown that neutrophils can promote the apoptosis of senescent blood vessels by releasing NETs, helping to eliminate pathological vessels [100]. Therefore, histone citrullination, through its regulation of NET formation, plays a pivotal role in the inflammatory response, microvascular damage, vascular remodeling, and regenerative repair processes in DR.

Histone citrullination can also influence DR through its impact on other histone modifications. Studies have demonstrated that PAD reduces histone H3 and H4 methylation by directly citrullinating arginine residues or converting monomethylated arginine to citrulline [101]. Moreover, PAD4 can activate HDAC1 and interact with HDAC2 and p53 through different domains, which synergistically repress transcription by promoting the deacetylation of histones [102].

Inhibiting PAD4 to reduce histone citrullination and prevent NET formation may emerge as a promising therapeutic approach for DR. Although PAD4 inhibitors have not yet been applied in ophthalmology, significant research has been conducted in other areas such as inflammation and cancer. For instance, Cl-amidine [103,104], BB-Cl-amidine [105], GSK199, and LDC7559 [106] have demonstrated potential therapeutic efficacy in NETosis-related diseases, with GSK199 and JBI-589 currently undergoing clinical trials for conditions like rheumatoid arthritis and cancer. These findings extend the scope of histone citrullination research in DR and offer new insights into potential therapeutic targets for DR treatment.

### 3.5. Histone Ubiquitination

Ubiquitin (Ub) is a highly conserved 76-amino acid protein found in eukaryotes. It plays a key role in histone modification by attaching to lysine residues on histones, thereby influencing their function and stability. While ubiquitination also occurs on histones H3 and H4, it is more prevalent on H2A and H2B, with H2A modification reaching up to 15% and H2B modification occurring at a rate of 1–2%. H2A ubiquitination (H2AK119ub1) is primarily catalyzed by the RING1A/B and BMI1 enzymes within the PRC1 complex, which is largely associated with gene silencing and chromatin compaction [107,108]. In contrast, H2B ubiquitination (H2BK120ub1) is catalyzed by the RNF20/RNF40 complex and plays a role in transcriptional activation and elongation [109]. Furthermore, H2B ubiquitination exhibits synergistic effects with histone methylation marks, such as H3K4me3 and H3K79me3, contributing to the regulation of gene expression [110].

Histone monoubiquitination occurs through a series of enzymatic steps. First, the ubiquitin-activating enzyme (E1) activates ubiquitin in an ATP-dependent manner [111,112]. Next, the ubiquitin-conjugating enzyme (E2) forms a thioester bond with ubiquitin via its cysteine residue [113]. Finally, a RING finger ubiquitin ligase (E3) transfers ubiquitin from E2 to specific lysine residues on histone substrates [114]. E3 ubiquitin ligases are crucial in this process as they recognize the histone substrates and catalyze the ubiquitination reaction. These modifications are integral to essential biological processes, including chromatin regulation, DNA repair, and transcription.

Studies have demonstrated a positive correlation between histone monoubiquitination levels and glucose concentrations [11]. Histone ubiquitination is a critical process in the DNA damage response, particularly in the repair of double-strand breaks (DSBs). Persistent DNA damage accumulation can accelerate neuronal aging, which is closely linked to the pathological progression of DR. Research has demonstrated that RAD6B, a key E2 ubiquitin ligase, facilitates DSB repair by ubiquitinating histone H2B. The loss of RAD6B function leads to increased DNA damage and retinal neurodegeneration, a phenomenon especially pronounced in aged mice [115]. This suggests that RAD6B is a vital regulatory factor in maintaining retinal homeostasis. Furthermore, RNF20, which localizes to DSB sites, recruits DNA repair proteins such as RAD51 and BRCA1 through H2B ubiquitination, participating in homologous recombination repair [116,117]. DR is an aging-related microvascular complication, with its progression strongly linked to DNA damage accumulation. These findings indicate that H2B histone ubiquitination plays a crucial role in DNA damage repair, and inactivation of the associated enzymes may contribute to the onset and progression of DR. Therefore, histone ubiquitination modifications could be central to the pathogenesis of DR.

### 3.6. Other Modifications

Histones can undergo SUMOylation, a post-translational modification in which a Small Ubiquitin-like Modifier (SUMO) protein is covalently attached to the lysine (K) side chains of histones. This process follows a mechanism similar to ubiquitination. Histone SUMOylation is closely linked to other modifications, such as histone deacetylation [118], potentially influencing gene expression through modification crosstalk and playing a role in the pathogenesis of DR. Studies have shown that the SUMOylation of H2A.Z at DNA damage sites is crucial for the DNA damage repair process. Rad52, a key recombination factor in the double-strand break (DSB) response, is a well-known SUMO substrate, although the precise role of SUMO modification of H2A.Z in DSB relocation remains unclear [119,120,121].

In addition to other modifications, all core histones and the linker histone H1 can undergo ADP-ribosylation. Histone mono-ADP-ribosylation is implicated in DNA damage and repair processes. Studies have shown that the type of ADP-ribosylated histones changes following DNA strand breaks. In the absence of DNA damage, mono-ADP-ribosylated histones are predominant, while levels of poly-ADP-ribosylated histones significantly increase after DNA strand breaks [122,123,124]. This suggests that the type of histone ADP-ribosylation could serve as a reliable biomarker for assessing DNA damage levels in DR patients, potentially helping to determine the severity of the disease. Additionally, H3R117 mono-ADP-ribosylation may promote cell proliferation by regulating P300 to increase cyclin D1 and c-myc levels, thereby influencing the neovascularization process in DR [125].

Histone phosphorylation is another key post-translational modification involved in DNA damage repair [126,127], cell cycle regulation, and chromatin remodeling [128]. Typically associated with active transcription, histone phosphorylation, like acetylation, facilitates chromatin relaxation by neutralizing the positive charge of histones. One of the first histone modifications induced by double-strand breaks (DSBs) is the phosphorylation of serine 139 on H2AX, forming γH2AX [129,130]. This modification plays a crucial role in recruiting DNA damage and repair proteins, initiating the DNA damage response. Additionally, research has shown that phosphorylation of H2BS14 is closely linked to chromatin fragmentation and apoptosis, making it an important marker for cell death signaling.

In addition to the more common histone modifications, there are several less prevalent types, such as crotonylation (Kcr), propionylation, and butyrylation. Studies have shown that levels of H3K9 crotonylation (H3K9cr) rapidly decrease following various types of DNA damage, suggesting that histone crotonylation plays a role in the DNA damage response (DDR) [131]. Furthermore, H3K9cr can promote the increased production of the inflammatory factor IL-1β, leading to IL-1β-dependent macrophage activation and exacerbating fibrotic lesions. Although research in ophthalmology is limited, these mechanisms of damage may similarly influence inflammatory responses and vascular pathology in DR.

## 4. Detection of Histone Modifications

Various methods are available for studying histone modifications, including protein immunoblotting (WB), immunohistochemistry (IHC), immunocytochemistry (ICC), and ELISA, all of which utilize specific antibodies to compare the overall levels of post-translational histone modifications across different samples. Mass spectrometry is employed to identify and characterize different modification types on histones. Co-immunoprecipitation (CoIP) and pulldown experiments are used to study protein interactions with endogenous histones and modification-dependent binding. High-throughput sequencing technologies, particularly next-generation sequencing (NGS), have significantly advanced protein research. Combining chromatin immunoprecipitation (ChIP) with NGS has enabled comprehensive mapping of epigenetic markers across the entire genome, utilizing techniques such as ChIP-seq, CUT&Tag, and nanoHiMe-seq.

Since its introduction by Barski et al. in 2007, ChIP-Seq has become a pivotal tool for studying epigenetic modifications, owing to its cost-effectiveness, efficiency, high sensitivity, and broad genome coverage [132]. In 2017, the Henikoff laboratory introduced the CUT&RUN technique to address challenges associated with traditional ChIP-Seq, such as high false-positive rates and poor antibody specificity [133]. CUT&RUN offers several advantages, including smaller sample requirements, a high signal-to-noise ratio, and lower cell demands. Then, the lab further advanced the field by launching the CUT&Tag technique, which leverages Tn5 transposase [134]. Unlike traditional ChIP-Seq methods, CUT&Tag eliminates the need for cross-linking, sonication, end polishing, and adapter ligation, providing benefits such as improved time efficiency, minimal sample requirements, reduced background noise, and high reproducibility [133]. The ongoing development of these techniques has greatly enhanced our understanding of epigenetic regulation, particularly histone modifications.

## 5. Conclusions

Histone modifications are a critical component of epigenetic regulation and play a pivotal role in understanding their functions and regulatory mechanisms in both physiological and pathological processes, offering valuable insights into disease development. While numerous studies have elucidated the involvement of various histone modifications in pathological changes associated with DR, such as alterations in cellular function, chronic inflammation, oxidative stress, fibrotic changes, and the DNA damage response (DDR), the mechanisms underlying many histone modification types in DR remain poorly understood. For instance, the roles of malonylation, 2-hydroxyisobutyrylation, and glutarylation require further investigation to fully elucidate their contributions to the disease.

Given the reversible nature of histone modifications, research in this area offers promising avenues for developing alternative preventive and therapeutic strategies for diabetic complications. Currently, the most well-established clinical therapies are histone deacetylase (HDAC) inhibitors, such as vorinostat, romidepsin, belinostat, and chidamide. These drugs inhibit histone deacetylation, promote chromatin relaxation, and enhance gene transcription, with the first three already being successfully applied in clinical treatments. Histone methyltransferase inhibitors, such as tazemetostat, have also shown clinical efficacy in anti-tumor therapy by suppressing abnormal histone methylation and regulating gene expression. Furthermore, histone acetyltransferase activators like curcumin, along with other drugs targeting histone modifications, are currently under clinical investigation. As epigenetic drugs, these therapies can target specific pathogenic genes and pathways, offering more precise and personalized treatments. While epigenetic drugs are predominantly used in cancer research, further studies focusing on DR are necessary to confirm their efficacy and potential adverse effects in this context. Drugs that modulate histones by regulating modifying enzymes may also have unintended consequences beyond the typical side effects of traditional DR treatments. Therefore, identifying more precise drug targets related to histone modifications is essential for improving therapeutic outcomes and minimizing potential adverse reactions.

## Figures and Tables

**Figure 1 biomolecules-15-00575-f001:**
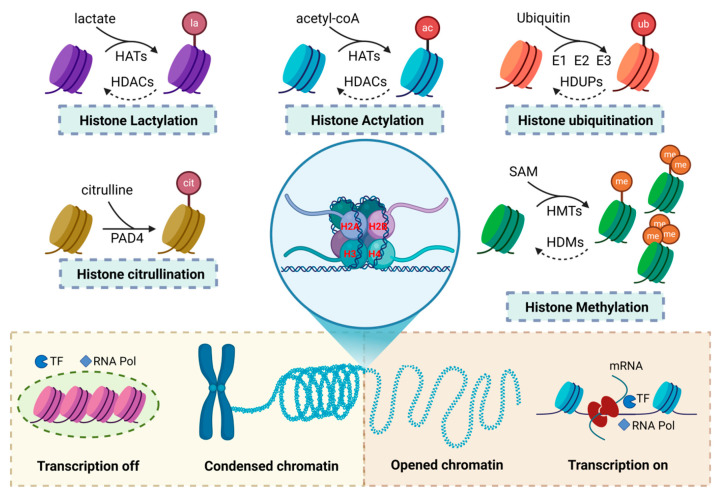
Types of histone modifications and their impact on chromatin structure and transcriptional regulation. This figure presents the common types of histone modifications and their effects on chromatin structure and transcriptional regulation. The modifications include lactylation, citrullination, acetylation, ubiquitination, and methylation. These modifications modify the chemical properties of histones by adding or removing specific chemical groups, which, in turn, influence the chromatin structure. Some histone modifications promote chromatin compaction, preventing the binding of transcription factors and RNA polymerase (RNA Pol), thereby facilitating transcriptional repression. In contrast, other modifications promote chromatin relaxation, enabling transcription factor and RNA polymerase binding to activate transcription.

**Figure 2 biomolecules-15-00575-f002:**
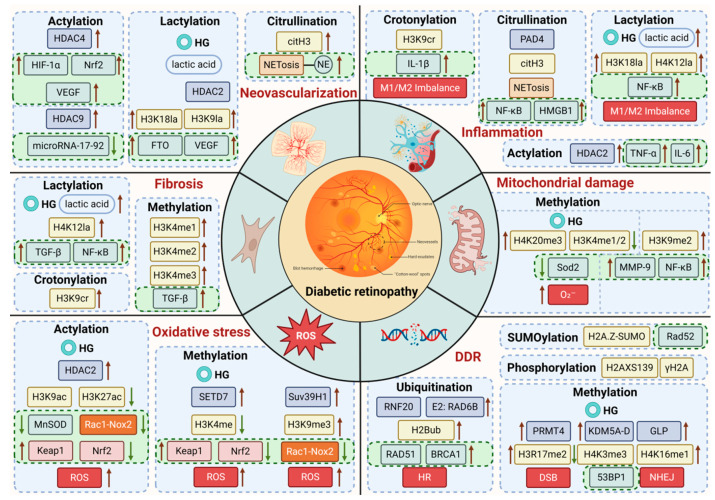
Mechanisms of high glucose and histone modifications in DR.

## Data Availability

Not applicable.

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
