# Peer review of "Histone Modifications: Potential Therapeutic Targets for Diabetic Retinopathy"

_biomolecules, 2025, doi:10.3390/biom15040575_

Round 1

Reviewer 1 Report (Previous Reviewer 2)

Comments and Suggestions for Authors

the new version is slightly improved but does not, in my opinion, offer any significant progress that would justify publishing it in a journal specifically dedicated to biomolecules.

Author Response

[April 1, 2025]

Dear Reviewer,

We would like to express our sincere gratitude for your meticulous review and constructive feedback on our manuscript. In response to your insightful comments on the novelty and contribution of our work, we have thoroughly revised and strengthened the manuscript. The key contributions of our paper are as follows:

This review, which focuses on histone modifications and diabetic retinopathy (DR), makes the following significant contributions:

  1. Filling the Gap in the Field

To date, no review has systematically synthesized the research on the relationship between histone modifications and diabetic retinopathy (DR). Our work is the first to provide a comprehensive overview of the advancements in this emerging field, addressing a significant gap in the current literature.

  1. Comprehensive Overview

We offer an in-depth synthesis of the latest advancements in the field of histone modifications, with particular emphasis on epigenetic regulatory mechanisms involved in DR. This provides researchers with a valuable and authoritative reference framework.

  1. Detailed Molecular Mechanism Analysis

We provide an innovative analysis of the specific regulatory networks of histone modifications, such as methylation and acetylation, in the pathogenesis and progression of DR, highlighting the pivotal role of epigenetics in the disease's pathological processes.

  1. Clinical Translation Potential

We conduct a thorough evaluation of the clinical applicability of targeted therapeutic strategies, such as HDAC inhibitors, offering novel ideas for intervention and new targets for drug development in DR treatment.

Once again, we deeply appreciate the invaluable comments and suggestions from the reviewers, and we hope the revised manuscript will meet your expectations.

Sincerely,

[Keran Li]

[The Affiliated Eye Hospital of Nanjing Medical University]

[138 Hanzhong Road, Nanjing]

[15852928023]

[025-86677679]

[likeran@njmu.edu.cn]

Reviewer 2 Report (Previous Reviewer 1)

Comments and Suggestions for Authors

The authors ha responded to the questions and criticisms raised by the reviewer. I appreciate how they have explained more in detail some of the points in the text, thus clarifying some of the issues. In my opinion the paper is now acceptable for publication.

Author Response

[April 1, 2025]

Dear Reviewer,

We would like to express our sincere gratitude for your recognition and positive evaluation of our revised manuscript. Your insightful comments and professional guidance throughout the review process have been immensely valuable. Your suggestions have significantly contributed to the improvement of the manuscript's quality, and your thorough review and constructive feedback have been crucial in enhancing the academic impact of this work.

We are delighted to hear that you consider the revised manuscript to meet the publication standards. Your endorsement is a great encouragement for our continued research. We remain committed to advancing our work in this field and to contributing further meaningful findings to the academic community.

Once again, we extend our heartfelt thanks for your invaluable feedback!

Sincerely,

[Keran Li]

[The Affiliated Eye Hospital of Nanjing Medical University]

[138 Hanzhong Road, Nanjing]

[15852928023]

[025-86677679]

This manuscript is a resubmission of an earlier submission. The following is a list of the peer review reports and author responses from that submission.

Round 1

Reviewer 1 Report

Comments and Suggestions for Authors

The paper by Lu and coworkers reviews the field of histone modifications in potential therapeutic targets for diabetic retinopathy.

After reading carefully the paper, there are several concerns which I think should be addressed. First, and most important is that this review should be of use for studies concerning diabetic retinopathy. Actually, the great majority of the papers a review of the literature on histone modifications and only the last paragraph deals with the clinical significance of histone PTM.

Other important points:

Abstract: not clear, with several repetitions and redundancy.

Table 1: The table is not clear. There are several cases in which Acetylation is indicated as ac, in others not. What is the difference between Transcript repression and Heterochromatin formation? In practice it is the same thing, since several PTM are present in both cases.

3.1 histone aceylation: RNA polymerase is not a transcriptional factor. The first paragraph is not clear, concerning the modification of charges. Moreover, on line 123 and following: it is not clear if hyperglicemia starts the acetylation process or causes sirtuuin inhibition.

3.4 histone lactation: not clear whether modifications in lactation leads to a positive or negative effect.

In synthesis, the paper is not sufficiently clear and balanced in its parts to be considered useful for a reader in the field.

Comments on the Quality of English Language

The English should be checked for repetitions.The are several typos in the text.

Author Response

Dear editors and reviewers:

Thanks for your crucial comments. The manuscript and response letter have been edited seriously, and the editors’ and the reviewers’ comments have been incorporated into the revised manuscript. The revised sections have been uploaded as an attachment. We hope that the current manuscript could meet the requirement for publication.

Thanks for your attention.

Sincerely,

Keran Li

Reviewer 2 Report

Comments and Suggestions for Authors

The article by Lu et al describes the link between histone modification and the development of diabetic retinopathy (DR).

This article, which would have been better placed in a medical journal, has the merit of examining a complex and interesting phenomenon, directly linked to the epigenetic modification of histones, a mechanism which in a sense concerns ‘biomolecules’ issues.

It is relatively well written and condensed, but I would suggest a number of improvements to make it easier to read.

General references on the state of the art concerning the structure of histones, nucleosomes and compact forms of chromatin are sorely lacking. Insofar as epigenetic modifications of histones affect chromatin compaction, a brief but detailed review of the subject is essential. DNA itself is modified by methylation at CpG sites. Is there a link with DRs?

So all in all, the introduction should be much better documented and referenced.

The models shown in Figure 2 are rather crude and don't seem to take into account recent advances in models of nucleosome compaction in chromatin fibres.

I also recommend making diagrams to explain the role of hyperglycaemia in the cascade of events leading to DR (described on page 5, line 123-144). The complex network of regulation would be better understood if it were represented by a figure.

I would also recommend, at the end, drawing a diagram summarising how all the different histone modification processes converge towards DR.

Author Response

(The authors gave the same response as above.)
